# Gender Trends in Healthcare and Academia—Where Does the University of Malta Stand?

Elizabeth Grech *, Anneka Pace, Tamara Attard Mallia and Sarah Cuschieri

Faculty of Medicine and Surgery, University of Malta, MSD 2080 Msida, Malta
* Correspondence: elizabeth.v.grech.18@um.edu.mt

**Abstract:** A current and pertinent topic is that of gender studies within healthcare students and academic staff of healthcare courses. This commentary explores the feminization of healthcare studies and the extent to which women in Malta hold key roles in academia within the faculties of Health Sciences, Dental Surgery, and Medicine and Surgery at the University of Malta. Data were publicly available from the university website. Gender (male: female ratio) trends were elicited from the data representing each level of qualification as offered by each faculty, while top academic roles within each faculty were noted. As a general trend, the number of students studying healthcare courses has increased, with an increased female-to-male ratio. Yet, in academia, men still occupy top roles. Efforts should be made to cater for all races, ethnicities, genders, sexual orientations, and socioeconomic levels within the healthcare workforce to allow delivery of the best possible service.

**Keywords:** gender equality; gender issues; gender bias; working women

## 1. Introduction

"Achiev[ing] gender equality and empower[ing] all women and girls" is one of the United Nations' Sustainable Development Goals (United Nations 2022). Identifying and addressing gender inequalities and gender equity is paramount to ensure good stewardship of health systems (Payne 2009). This commentary sheds light on gender trends within academia and health studies at the University of Malta. There is only one state university in Malta where both local and international students are enrolled. We explore the feminization of healthcare studies and the extent to which women in Malta hold key roles in academia within the faculties of Health Sciences, Dental Surgery, and Medicine and Surgery. Understanding the situation in Malta's only state university is therefore of key interest to gender studies in healthcare and academia on a global scale.

This commentary is an observational piece that provides a representative overview of the gender trends in academia at university level within the country and may be compared with other international universities. We describe trends in healthcare courses across the years, starting from 2009 and continuing up to 2022, as well as the current gender imbalances (academic year 2022–2023) within different levels of academic staff.

Understanding gender trends in universities and the possible driving forces for these trends is essential as it allows human resource planning and may stimulate the development of policies or benefits that suit the needs of the workforce, both in health academia and healthcare itself (El Arnaout et al. 2019).

Our commentary aims to understand the gender trends in Malta's university so that our findings may be applied to the wider international scientific community. This would allow a better understanding of how universities contribute to the attainment of Sustainable Development Goals in terms of gender equality and women's rights.

## 2. Data Collection—Staff and Students

All data used were publicly and freely available and were taken from the University of Malta website. Data pertaining to students included the list of available courses for the

faculties of Health Sciences, Dental Surgery, and Doctor of Medicine and Surgery for each academic year, and the number of male and female students admitted to each course yearly. Data relating to staff were found by accessing the website of each faculty and inputting the gender of the dean, deputy dean, and heads of department into an Excel spreadsheet. These represent the top academic roles within each faculty.

Gender (male: female ratio) trends were elicited from the data representing each level of qualification as offered by each faculty. Data for academic staff were organized into a table.

### 2.1. Gender Trends: Students in Healthcare

Female enrolment increased relative to that of males within the Faculty of Dental Surgery over time across all educational levels. In contrast, diploma-level courses have been consistently dominated by the female gender. This can be seen in Table S1.

Diplomas and Bachelors qualifications are dominated by female students within the Faculty of Health Sciences. However, males appear to be more inclined to undergo post-graduate courses such as Masters and PhDs. This is illustrated in Table S3.

Vertical and horizontal occupational segregation is present when looking at healthcare positions and gender. Women tend to be underrepresented when it comes to managerial and decision-making roles, with females in the healthcare workforce usually concentrated in professions to do with "care" such as nursing and midwifery. These occupations tend to be perceived as "low-status" jobs when compared to medicine, dentistry, and pharmacy. These "high-status" occupations tend to be occupied by men (European Institute for Gender Equality 2019). The trends seen within the faculties of Dental Surgery and Health Sciences echo this observation.

The course of Doctor of Medicine and Surgery within the Faculty of Medicine and Surgery is one of the largest courses, in terms of the number of students, that exists within the University of Malta. It shows an interesting trend in that male dominance existed until 2013. From then onwards, the number of males to females remained virtually equal, with the only exception being the 2021–2022 academic year, where there were more females than males. Conversely, a female predominance was observed across all academic years in post-graduate courses, i.e., Masters and PhDs, within the Faculty of Medicine and Surgery. This can be seen in Table S5.

The feminization of the medical course in Malta follows global trends where data show more women have entered medical school compared to men in the past two decades. There is evidence that the growing global feminist movement, in which women triumphed over prejudices and sexism, has helped women to gain their place within this labor market (Perinni 2021). Participation of women in the medical profession has in fact been shown to have increased over the past four decades, as can clearly be seen within the University of Malta (Jagsi et al. 2006).

Census data from Canada and the United States are congruent with Maltese findings in that the feminization of women in healthcare professions is clearly evident (Adams 2010). A greater proportion of female physicians compared to males was also demonstrated in Bangladesh in 2019. Student admission data also showed a female majority (Hossain et al. 2019). Although Japan has a relatively low proportion of female physicians, female participation in the medical field has been shown to have increased over the past decade, a trend that matches that in Europe, including Malta (Nomura et al. 2010). The proportion of Scandinavian women in medicine relative to males has surpassed that of the United States, although underrepresentation of the female sex in higher administrative positions exists in Finland (Riska 2001). Soviet history resulted in the feminization of the medical profession in Russia earlier than in the West; however, studies have demonstrated that only a small number of women are found in prestigious specialties and in academic medicine (Harden 2016; Ramakrishnan et al. 2014; Riska 2001).

It is a well-known fact that medical teams with greater gender diversity among senior roles perform better (European Union of Medical Specialties 2020). Yet, it appears that this

is still not reflected in undergraduate enrollment choices, especially among allied healthcare students, which will make up the next generation of workers. Strides towards achieving this should thus be made on a global scale and would require worldwide collaboration of institutions involved. Dealing with phenomena such as the gender pay gap, where women in the health sector earn 20% less than their male counterparts, would encourage women to join the profession and decrease turnover rates (United Nations 2022). Although the past decade has seen an increase in the female medical workforce, they occupy most of the lower-status and lower-paid jobs compared to men. International collaboration is required to develop a health care system that recognizes and celebrates the contributions of the female sex (World Health Organization 2022).

### 2.2. Gender within Academia—A Snapshot

A general male predominance is present within the positions of dean and deputy dean for these faculties. When it comes to heads of department, a significantly larger proportion within the Faculty of Health Sciences are female while an equal number of males and females are present in these roles in the Faculty of Medicine and Surgery. A female predominance in heads of department exists in the Faculty of Dental Surgery. This is shown in Tables S2, S4 and S6.

The field of medicine and surgery is one that has traditionally been commanded by men, and more men tend to occupy more senior positions. This occurs in almost all departments, but more so in the field of surgery and anesthesia. The issue of lack of opportunities that prevents women from continuing their education is one that should be explored worldwide (Ayomi 2021). Women remain significantly underrepresented in the roles of senior doctors and full professors on a global scale (Kuhlmann et al. 2017). This is concordant with the observations within this commentary as although many women are indeed heads of department, the more senior roles of dean and deputy dean remain male-centric. Women in academic healthcare fields experience inequity because of sexism and other issues, such as carrying a greater burden of domestic responsibilities and the need for caregiving leave and facilities. As a result, it is generally difficult to keep up with their male colleagues (Carr et al. 2018). Globally cooperative strategies and responses are needed to counteract this issue (Morgan et al. 2018). These may include, but are certainly not limited to, childcare provision, ensuring fair and equal pay, and better representation for women at all levels. Such strategies will provide women with more autonomy over the management of their lives and will allow women to progress in their career while also balancing domestic duties (Berlin et al. 2019; Freund et al. 2016; MacDonald 2003). These measures serve to overcome challenges brought about by women's historical and social disadvantages that prevented them from operating on a level playing field alongside men.

In conclusion, parity for all healthcare workers is essential for productivity and the provision of a good service (Valantine 2020). Given that healthcare workers treat diverse patients, having a diverse workforce as a reflection of that would allow delivery of the best care possible. Efforts should be made to cater for all races, ethnicities, genders, sexual orientations, and socioeconomic levels within the healthcare workforce to allow delivery of the best possible service through a workforce that is a reflection of the diverse patients that it caters for (Stanford 2020).

**Supplementary Materials:** The following supporting information can be downloaded at: https://www.mdpi.com/article/10.3390/socsci11100463/s1, Table S1. Students within the Faculty of Dental Surgery across 15 academic years; Table S2. Academic Staff within the Faculty of Dental Surgery for academic year 2022/2023.; Table S3. Students within the Faculty of Health Sciences across 15 academic years.; Table S4. Academic Staff within the Faculty of Health Sciences for academic year 2022/2023. Table S5. Students within the Faculty of Medicine and Surgery across 15 academic years. Table S6. Academic Staff within the Faculty of Medicine and Surgery for academic year 2022/2023.

**Author Contributions:** Conceptualization, S.C.; methodology, S.C. and E.G.; software, S.C.; validation, S.C. and E.G.; formal analysis, S.C.; investigation, S.C., E.G., A.P. and T.A.M.; resources, S.C.; data curation, S.C., E.G., A.P. and T.A.M.; writing—original draft preparation, S.C. and E.G.; writing—review and editing, S.C., E.G., A.P. and T.A.M.; visualization, S.C., E.G., A.P. and T.A.M.; supervision, S.C.; project administration, S.C. All authors have read and agreed to the published version of the manuscript.

**Funding:** This research received no external funding.

**Institutional Review Board Statement:** Not applicable.

**Informed Consent Statement:** Not applicable.

**Data Availability Statement:** Data used was publicly and freely available online from the University of Malta Website: https://www.um.edu.mt/.

**Conflicts of Interest:** The authors declare no conflict of interest.

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
