# Peer review of "Gender Trends in Healthcare and Academia—Where Does the University of Malta Stand?"

_socsci, doi:10.3390/socsci11100463_

Round 1
Reviewer 1 Report
The commentary focuses on a relevant issue in current social research: the gender gap in academic institutions. It comments on the presence of women in academia and health studies at the University of Malta, particularly the vertical segregation that manifests itself in the difficulty of women in career progression and in reaching the top and more prestigious positions.
The data confirm the trend found in other European countries. A comparative analysis with data from different disciplines and countries could give more excellent value to the commentary. The literature cited could be expanded by recalling the numerous studies conducted in recent decades on the gender balance of universities and the mechanisms and processes underlying gender segregation in scientific institutions.
Author Response
The commentary focuses on a relevant issue in current social research: the gender gap in academic institutions. It comments on the presence of women in academia and health studies at the University of Malta, particularly the vertical segregation that manifests itself in the difficulty of women in career progression and in reaching the top and more prestigious positions.
Thank you for your comment, this was indeed what we aimed to shed light on.
The data confirm the trend found in other European countries. A comparative analysis with data from different disciplines and countries could give more excellent value to the commentary. The literature cited could be expanded by recalling the numerous studies conducted in recent decades on the gender balance of universities and the mechanisms and processes underlying gender segregation in scientific institutions.
Thank you for your comment. We have added a comparative analysis with data from Canada, the United States, Bangladesh, Japan, Scandinavia and Russia. We also addressed some of the mechanisms and processes underlying gender segregation in universities while keeping the manuscript succinct, as per author guidelines.
Reviewer 2 Report
Gender trends in healthcare and academia – where does the University of Malta stand?
General conclusions: The paper is not a scientific article but it is rather a summary of the article. There is no scientific goal presented in the article, no research questions or hypothesis to verify. To little attention is paid on the theoretical discourse. It should be shown that there are some researches on the subject and what is the scientific contribution of the presented article in the discourse.
In the parts 3 and 4 (lines 49 and further lines) the Authors presents some research findings but without presenting any data. There are no tables with numbers of student or employees – again it is rather a conclusion part not a research part of the article.
Author Response
General conclusions: The paper is not a scientific article but it is rather a summary of the article. There is no scientific goal presented in the article, no research questions or hypothesis to verify.
Thank you for your comment. Given that the article is a commentary and not a full research article, we did not include any specific scientific goals or hypothesis. We have however adapted a paragraph in the introduction to better detail our research question and highlight the aim of this piece.
To little attention is paid on the theoretical discourse. It should be shown that there are some researches on the subject and what is the scientific contribution of the presented article in the discourse.
Thank you for your comment. We have added a comparative analysis with data from Canada, the United States, Bangladesh, Japan, Scandinavia and Russia in order to enhance the theoretical discourse of the article. We also addressed some of the mechanisms and processes underlying gender segregation in universities while keeping the manuscript succinct, as per author guidelines.
We have also adapted the introduction to make the scientific contribution of the article clearer.
In the parts 3 and 4 (lines 49 and further lines) the Authors presents some research findings but without presenting any data. There are no tables with numbers of student or employees – again it is rather a conclusion part not a research part of the article.
Thank you for pointing this out. We have added a table for each faculty to better illustrate our data, however, given that this is a commentary and not a full research article we feel it would be beyond the scope of this piece to go into further detail.
Round 2
Reviewer 2 Report
no more suggestions